# A Review of Traumatic Brain Injury and the Gut Microbiome: Insights into Novel Mechanisms of Secondary Brain Injury and Promising Targets for Neuroprotection

**DOI:** 10.3390/brainsci8060113

**Published:** 2018-06-19

**Authors:** Caroline S. Zhu, Ramesh Grandhi, Thomas Tyler Patterson, Susannah E. Nicholson

**Affiliations:** 1Division of Trauma and Emergency Surgery, Department of Surgery, The University of Texas Health Science Center at San Antonio, 7703 Floyd Curl Drive (MC 7740), San Antonio, TX 78229, USA; zhuc5@uthscsa.edu (C.S.Z.); ramesh.grandhi@hsc.utah.edu (R.G.); PattersonTT@livemail.uthscsa.edu (T.T.P); 2Department of Neurosurgery, The University of Texas Health Sciences Center at San Antonio, 7703 Floyd Curl Drive, San Antonio, TX 78229, USA; 3Department of Neurosurgery, The University of Utah School of Medicine, Salt Lake City, UT 84132, USA

**Keywords:** Traumatic brain injury (TBI), microbiome, gastrointestinal (GI), gut, commensals, brain-gut axis, central nervous system (CNS), enteric nervous system (ENS)

## Abstract

The gut microbiome and its role in health and disease have recently been major focus areas of research. In this review, we summarize the different ways in which the gut microbiome interacts with the rest of the body, with focus areas on its relationships with immunity, the brain, and injury. The gut–brain axis, a communication network linking together the central and enteric nervous systems, represents a key bidirectional pathway with feed-forward and feedback mechanisms. The gut microbiota has a central role in this pathway and is significantly altered following injury, leading to a pro-inflammatory state within the central nervous system (CNS). Herein, we examine traumatic brain injury (TBI) in relation to this axis and explore potential interventions, which may serve as targets for improving clinical outcomes and preventing secondary brain injury.

## 1. Introduction

Recent estimates of the magnitude of traumatic brain injury (TBI) suggest that each year, approximately 1.7 million people sustain a head injury in the United States [1]. TBI is the leading cause of death under the age 45 in the Western World and is a contributing factor in 30.5% of all injury-related deaths that occur in the United States, with direct medical expenditures and indirect costs attributable to TBI upwards of $60 billion in the year 2000 [2].

Physiological effects following TBI have been increasingly studied, with intestinal dysfunction representing an important consequence [3,4,5,6]. The disruption of the brain-gut axis, the major bidirectional communication pathway between the brain and the gastrointestinal system, which incorporates both afferent and efferent signals involving neuronal, hormonal, and immunologic pathways, can result in sequelae such as chronic dysfunction of the gastrointestinal system and disability [7,8].

Study of the human biome, a complex ecosystem comprised of upwards of 3 × 10^13^ bacterial cells that outnumber the quantity of cells within the human body, represents a novel interest in medicine. With respect to the gut, the colon is the most colonized area within the human body [9,10]. Noting previous research that associated changes in the gut microbiome with autism, major depressive disorder, and Guillain–Barre syndrome, Ochoa-Reparaz and colleagues discussed in their animal study that modifications of the gut microbiome could result in dysregulation of immune responses in the central nervous system (CNS). In turn, they postulated that changes in the gut microbiota would potentiate autoimmune processes or, alternatively, protect against proinflammatory conditions in the CNS [11]. A pre-clinical study by Houlden et al. found changes in the gut microbiota of mice following experimental stroke and traumatic brain injury, thereby demonstrating that brain function and the intestinal microbiome are related in a type of feedback loop in which each component affects the other [12]. Additionally, our group has recently shown that significant changes in the gut microbiome are evident within two hours following a moderate TBI induced in rats with loss of α-diversity and dysbiosis occurring. Moreover, phylogenetic changes and α-diversity were significantly correlated with MRI-determined lesion volume [13]. In a recent systematic review, Brenner et al. examined the few clinical studies conducted thus far on prebiotic and probiotic interventions for those suffering from TBI and/or posttraumatic stress disorder. The review found that in the three clinical studies involving TBI patients, a probiotic or prebiotic course of action improved patient health compared to control groups [14].

Given the influence of the composition of the gut microbiome on the homeostasis of the CNS, the purpose of this manuscript is to review the literature published on the gut microbiome and its importance in immunity and injury. Furthermore, we discuss the dynamic relationship between the brain-gut axis and brain injury, specifically TBI. By developing an understanding of the role that the brain-gut axis has in potentiating inflammation or protecting against secondary injury, we can enhance patient care, identify therapeutic targets for neuroprotection, reduce morbidity, and improve outcomes.

## 2. Gut Microbiome and Its Role in Immunity

Composed of trillions of bacteria, the gut microbiome is crucial to human immunity [15]. The gastrointestinal tract consists of about 10^13^–10^14^ microorganisms from over 1000 different species, outnumbering the cells in our bodies. The bacterial phyla *Bacteroidetes* and *Firmicutes* primarily define the microbiome, with *Proteobacteria*, *Actinobacteria*, *Fusobacteria*, and *Verrucomicrobia* in lesser prevalence [16]. The composition of the microbiome varies from person to person, with contributing factors including age, diet, behavior, environment, and genetics [17].

The gut microbiome serves numerous functions in the human body and is so crucial to our survival that it has been dubbed “our forgotten organ.” Some of the roles it serves include digestion of polysaccharides, development of the immune system, defense against infections, regulation of angiogenesis, and production of essential proteins that our genes do not encode [18]. The gut microbiota also establishes the intestinal barrier, assists with mucus production, and promotes regeneration of intestinal epithelial cells. In addition, the microorganisms in the gut stimulate the innate immune system, leading to a cascade of events that culminate in immune system maturation. Furthermore, the flora in our gastrointestinal tract continuously stimulate our immune system by triggering a low level of inflammation, which in turn helps our bodies defend against harmful microbes [19].

The intestinal epithelium facilitates nutrient absorption, electrolyte balance, and provides a mechanical barrier to prevent commensal organisms from entering and to protect against invading pathogens [20,21]. It also coordinates an immune response that activates host defenses. Protective mechanisms of the intestine include mechanical defenses such as the presence of tight junctions and the epithelial barrier itself, as well as peristalsis, and production of a mucus layer [20,21]. Non-mechanical defenses include colonization by commensals, secretion of immunoglobulins such as IgA, and the presence of a highly specialized network of dendritic cells that sample the intestinal lumen and transport luminal antigens to lymph nodes, as well as macrophages and antigen receptors such as toll-like receptors (TLRs) [21,22]. TLRs are integral in the activation of the innate immune response, and are mainly expressed by cells in tissue with immune function such as the spleen and peripheral blood leukocyte [19]. They are also found in tissue that is exposed to the environment such as in the lung and in the GI system [19,22].

Evidence from multiple pre-clinical studies suggests that commensal gut microbiota affect the intestinal immune response. Commensals influence gut-associated lymphatic tissue (GALT) formation, induction of Peyer’s patches with induction of mucosal T cells and IgA plasma cells. The microbiota can induce pro-inflammatory and anti-inflammatory responses. Germ-free mice have been shown to have decreased cellularity and functionality of the small intestine immune system with fewer plasma cell and intra-epithelial lymphocytes, lower IgA levels, smaller Peyer’s patches, decreased lamina propria CD4+ and intra-epithelial CD8αβ+ cells and increased susceptibility to pathogenic bacteria [20,22,23,24,25]. Germ-free mice also lack Th17 cells, a subset of CD4+ T cells that secrete IL-17, which are a prominent T cell population found in the intestinal lamina propria that help maintain intestinal homeostasis by improving barrier function, stimulating mucin production, affecting tight junction functionality and IgA transport [20].

Numerous studies involving mouse models have demonstrated a key role of TLRs in the immune system of the gastrointestinal tract [26,27,28,29]. Intestinal epithelial cells express pattern recognition receptors (PRRs) including TLRs [22]. Bacteria possess pathogen-associated molecular patterns (PAMPs) that can interact with PRRs and induce the innate immune response. Commensal components are recognized by TLRs, and impairment in the interaction between commensal organisms and TLRs has been shown to promote chronic inflammation and tissue damage [20]. TLR2 and TLR5 are regulated by microbiota in the colon. Epithelial cells in germ-free mice have decreased TLR expression, suggesting that the microbiome is involved in PRR expression [25]. The interaction between MyD88, a protein involved in many inflammatory pathways, inclusive of IL-1 and TLR signaling pathways in the innate immune system, and TLRs in gut T cells coordinates germinal center responses including T follicular helper (T_FH_) cell, and thus IgA secretion and B cell development. T_FH_ development is deficient in germ-free mice but restored by feeding TLR2 agonists [23,30]. Given the prominent role of the gut and its microbiome in relationship with the immune system, it is unsurprising that distortion of microbial homeostasis has been associated with many diseases and conditions. Multiple clinical and pre-clinical studies have linked the gut microbiome to disorders such as obesity, demyelinating disease, and cardiovascular disease [31,32,33,34,35,36,37]. Many inflammatory diseases, including rheumatoid arthritis and inflammatory bowel disease have also been correlated with an altered gut microbiome [38,39,40]. For instance, a recent clinical study has revealed that patients suffering from psoriasis, an inflammatory autoimmune condition, display a very different gut microbiome compared to healthy individuals—every psoriatic patient in this study had an increase in *Faecalibacterium* and a decrease in *Bacteroidetes* [41]. Several clinical studies have even shown that changes in the gut microbiota participate in the development of non-alcoholic fatty liver disease and type 1 diabetes [32,42,43,44]. In light of our developing understanding of the role of the gastrointestinal microbiota in an array of inflammatory conditions and diseases, there is abundant opportunity for future therapies to target the gut microbiome.

## 3. The Gut Microbiome and Systemic Injury

Changes to the gastrointestinal environment from trauma, burns, sepsis, and surgical injuries trigger bacteria dysbiosis in the gut, which leads to the development of a systemic inflammatory response [45]. This type of response may be due to a catecholamine surge, which in turn leads to problems on the local level. Several groups have found that autonomic nervous system dysregulation and stress response after stroke mediate the relationship between acute brain injury and remote organ dysfunction [46].

Earley et al. found that burn injury changed the composition of the gastrointestinal microbiome with an overgrowth of many previously less common taxa along with concomitant reduction of healthy bacterial diversity, leading to the overabundance of Gram-negative aerobic bacteria [47]. The authors also noted important changes to structure resulting in increased intestinal permeability, which coupled with the overgrowth of *Enterobacteriaceae*, allowed for translocation of these bacteria to the mesenteric lymph nodes. These findings, in turn, suggest that the gut epithelium itself is implicated in bacterial infections and/or sepsis following burn injury [47]. The combination of alcohol and burn injury has been separately shown to lead to dysregulated mucin production and tight junction expression with increases in *Enterobacteriaceae* in the small and large intestines [48].

In regards to traumatic injury, Hayakawa and colleagues found that the intestinal microbiota becomes fundamentally disrupted within hours of injury and sudden physiologic insult including cardiac arrest and stroke [49]. Similarly, a recent study in severely injured patients demonstrated significant changes in phylogenetic composition and relative abundance in the first 72 h following injury [50]. Critically ill patients admitted with the systemic inflammatory response (SIRS), including those with traumatic injury, have also been shown to have alterations in the gut microbiome, and these disturbances are predictive of sepsis-associated mortality in their patient population. Another instance of surgical injury altering the gut microbiome was described by Shogan et al., in their study revealing that intestinal anastomotic injury can provoke changes in the gut microbiome [51]. Furthermore, amongst critically ill patients with a prolonged hospitalization, the composition of the gut microbiome dramatically changes such that pathogen communities of extremely low diversity emerge and trigger further virulence in the host [52]. Similarly, critical illness can also cause a shift in the microbiome in such a way that dysbiosis occurs in which beneficial microbes are lost and pathogenic bacteria begin to monopolize the gut [53].

Increased intestinal inflammation and reduced antimicrobial peptides appear to exert a key influence in the pathophysiologic processes following injury [47]. Intestinal wall damage leads to inflammation of the mucosal barrier, which results in an altered GI tract with higher levels of nitrate and abnormal gradient of mucosal oxygen [54,55,56]. As a result, the changes to the environment lead to a proliferation of *Proteobacteria* such as *Pseudomonas aeruginosa* and *Escherichia coli*, as well as certain *Firmicutes* including *Staphylococcus aureus* and *Enterococcus* spp. [56,57,58,59]. Thus, the dangerously low diversity and instability of the microbiome ecosystem more closely resembles that of an infectious state [56]. Krezalek et al. have further suggested that the microbiome shifts to a “pathobiome” as a result of sepsis and surgical injury. This pathobiome is a culmination of intestinal microbes that become pathogenic due to insult to the host body and can ultimately result in death of the host [60].

Intestinal ischemia-reperfusion has also been implicated in the pathophysiology of changes to the gut microbiome. Damage to the epithelial barrier of the colon directly follows intestinal ischemia-reperfusion and leads to dysbiosis of the gut microbiome. *Escherichia coli*, *Prevotella oralis*, and *Lactobacilli* were found to flourish and induce gut inflammation among other detrimental effects. In particular, *Escherichia coli* appear to play a central role by effecting changes in tight junction proteins in the intestinal epithelial barrier [61]. These pathophysiologic changes in the gut microbiome then, through various communication pathways such as the brain-gut axis, feed back to the brain and result in changes in the CNS.

## 4. Brain-Gut Axis and Neurologic Injury

The brain-gut axis (BGA), also referred to as the gut-brain axis, is a communication network that links together the CNS and the enteric nervous system (ENS) [21]. This axis functions in a bidirectional manner; thus, each component of the network can influence the other. The downstream communication from the brain to the gut involves vagal pathways that innervate the ENS, which is critical for the function of the GI tract [62,63,64]. From a gut-to-brain viewpoint, numerous factors such as gut lipopolysaccharides, cytokines, neuropeptides, and protein messengers connect the gastrointestinal tract to the brain [65].

With the advances in our understanding of the significance of the gut microbiome, mention of the brain-gut axis in more recent literature often includes the microbiota, with some researchers renaming it as the brain-gut-enteric microbiota axis [16]. The gut microbiome is critical to the development of the nervous system [66]. The gastrointestinal microbiota influences the ENS, which in turn sends signals to the CNS [67]. Gut microbes also regulate glial cells in the intestines, which are essential in connecting the ENS to the CNS [68]. Wang et al. noted that the gut microbiota influences the brain through a number of pathways, including through the nervous, endocrine, immune, and metabolic systems. The gut microbiome can affect the brain via one of many different pathways—the neuroanatomical pathway of the gut-brain axis, the neuroendocrine–hypothalamic–pituitary–adrenal axis, the gut immune system, the gut microbiota metabolism system, and the intestinal mucosal barrier and the blood brain barrier [8]. In addition, intestinal health affects the CNS via the physiological effects of intestinal barrier function and peripheral neuron activity [69]. Being bidirectional, the communication pathways also extend downstream from the brain to the gut, with changes in neurophysiological behaviors impacting the gut microbiome [66].

Disturbances of the BGA are implicated in many neurologic disease processes. Furthermore, the relatively recent discovery of the significance of the gut microbiota to the brain-gut axis has led to the reevaluation of CNS diseases [70]. The gut microbiota likely plays a role in a wide range of neurological conditions, including autism spectrum disorder, anxiety, depression, chronic pain, stress, Alzheimer’s disease, and Parkinson’s disease [66,70,71]. In fact, intestinal inflammation and pathology in Parkinson’s disease precedes pathologic changes observed in the CNS by decades [69]. The gut microbiome has been suggested to both contribute to the onset of these neurological conditions as well as influence the severity of these disorders [72]. In addition, given its influence on pharmacokinetic properties, the gut microbiome can also play an important role in the effectiveness or side effects of medications used to treat these diseases [73].

Changes in the gut can also drive progression of stress-related disorders within the CNS through the BGA [74]. Pre-clinical models have also demonstrated a relationship between gastrointestinal dysfunction and autism spectrum disorders, with a focus on the role of the BGA [75]. In addition, several clinical studies have examined the relationship between the gut microbiome and brain activity [76]. Pinto-Sanchez et al. studied patients suffering from irritable bowel syndrome (IBS) and discovered that a probiotic treatment effectively reduced depression and improved quality of life in IBS patients [77]. Another clinical study focused on anorexia nervosa and documented the association between decreased bacterial diversity and increased levels of depression and anxiety [62]. Nagy-Szakal et al. also linked intestinal dysbiosis with myalgic encephalomyelitis/chronic fatigue syndrome [78]. Gastrointestinal symptoms were even implicated with autism spectrum disorder in a multicenter clinical study [79].

Other examples of BGA disturbances and CNS illness are of autoimmune or inflammatory origin. Berer et al. identified a potential pathway in which the commensal gut microbiome is a key player in activating autoimmune demyelination in mice [80]. A separate animal study by Lee and colleagues revealed that the intestinal microbiota regulates pro- and anti-inflammatory responses in the gut and in the CNS. More specifically, Lee et al.’s study showed that signals originating from the gut microbiome can influence the immune system via the Th1/Th17 vs. Treg axis and trigger inflammation beyond the intestine in the CNS. They also found that the gut microbiota is instrumental in the induction of experimental autoimmune encephalomyelitis, an experimental model of multiple sclerosis [81]. It has also been suggested that disruption of the BGA could be a factor in inflammatory bowel disorders [63].

Damage to the intestinal barrier and changes in the gut microbiome have been seen following stroke as well. Using a murine middle cerebral artery stroke model, Singh et al. observed a significant decrement in the diversity of bacteria composing the gut microbiome in the days following stroke. Other findings include reduced gastrointestinal motility, leading to overgrowth of bacteria, and a functionally impaired intestinal barrier. In addition, the authors noted that bacterial dysbiosis led to translocation of peripherally activated T cells to the brain, resulting in an inflammatory cascade that ultimately resulted in increased stroke volumes and worse stroke outcomes [46]. The significant role of intestinal microbiota in influencing neuroinflammation as mediated by T cells following stroke was also shown by Benakis et al., suggesting that influencing the composition of the gut microbiome could potentially serve as a therapeutic target [82].

Specific to traumatic brain injury, disruption of the brain-gut axis can trigger subsequent complications. Amongst adult patients with TBI, 8–33% of individuals exhibit autonomic imbalance [83,84,85]. TBI-induced disruption of the corticopontine connections affecting the vagus complex can result in dysautonomia, which appears to have a significant effect via the BGA. As seen following stroke, the release of noradrenaline triggers changes in the gut microbiome, which may clinically manifest in patients as abdominal pain, gastric distension, altered intestinal motility, constipation, or ulcers [12]. Similarly, TBI patients with dysautonomia sustain more complications and experience worse outcomes [82]. The disturbance of this axis between the gut and vagus complex in brain-injured patients can also result in physiologic intestinal changes, which likely also impact the gut microbiome [86,87].

TBI also causes increased intestinal permeability secondary to decreased expression of intestinal tight junction proteins ZO-1 and occludin [3]. The consequent disruption of the anatomic and functional integrity of the gut can result in systemic inflammation, bacterial translocation, and sepsis. In addition, Houlden et al. showed that TBI affects the composition of gut microbiota in the caecum and drew a link between TBI severity and changes in the microbiome, specifically, in the composition of *Bacteroidetes*, *Porphyromonadaceae*, *Firmicutes*, and *Proteobacteria*. The authors hypothesized that these observations may be a result of stress response due to tissue injury, as well as secondary to alterations in brain function. Additionally, these changes also correlated with increased levels of noradrenaline, a known factor in the induction of microbiome dysbiosis [12]. Findings from our laboratory demonstrated that the gut microbiome is altered within two hours following a moderate TBI in rats, with varying trends among the phylogenetic families and loss of species diversity in the absence of other injury patterns or therapeutic intervention. Phylogenetic changes and dysbiosis persisted throughout 7 days. A decrease in relative abundance in traditionally beneficial bacteria was observed, specifically in the families *Lachnospiraceae*, *Mogibacteriaceae*, and *Ruminococcaceae* within the phylum *Firmicutes*. Conversely, families that contain pathogenic bacteria including *Bacteroidaceae* in the phylum *Bacteroidetes*, and *Enterobacteriaceae* and *Pseudomonadaceae* in the phylum *Proteobacteria* were increased. Many of the microbial alterations were observed at three days post-TBI and correlated with peak MRI lesion volume and loss of behavioral function. Moreover, we found that a larger brain lesion was associated with greater decreases in levels of *Firmicutes*, an exacerbated increase in *Proteobacteria*, and a significant reduction in α-diversity [13]. While the exact mechanism remains unknown, a feedback loop initiated by TBI with resultant increased pathogenic flora may influence the BGA and potentiate a neuroinflammatory cascade, which then leads to secondary brain injury and influences functional outcome. The death of neuronal cells, ischemia, hemorrhage, and blood brain barrier (BBB) disruption that occurs secondary to TBI may also initially trigger changes at the gut level by affecting the BGA by activating pro-inflammatory cytokine cascades and by impacting enteric glial cell and regulatory T cell activation and differentiation [88,89]. Of particular interest is the suppression of T_reg_ cell differentiation, which may have a profound effect on subsequent intestinal changes and neuroinflammation [82]. The physiologic and clinical implications of TBI-induced changes to the gut microbiome remain unknown. Further investigation of the gut microbiome following TBI has the potential to improve detection of TBI and outcome and serve as a potential therapeutic target.

## 5. Potential Therapeutic Interventions

Investigation into the BGA in the setting of systemic injury and TBI has identified several promising targets for intervention. One possible treatment involves mitigating the gut dysbiosis that results from TBI by attempting to restore the normal gut microbiota. Fecal microbiota transplant (FMT) is one method of addressing this problem and involves taking fecal matter from a donor, mixing it with a solution, and placing the strained fecal solution into a patient to replace the lost beneficial bacteria [90,91]. FMT has been used to successfully treat several conditions including *Clostridium difficile* infection, irritable bowel syndrome, Crohn’s disease, and ulcerative colitis [91].

Although not thoroughly explored for TBI, probiotics may offer another potential therapeutic option by increasing IL-10 production and decreasing intestinal epithelial cell production of pro-inflammatory cytokines [92,93]. In addition, probiotics have also been found to reduce intestinal permeability through modulating the hypothalamic-pituitary-adrenal (HPA) axis. A study by Ait-Belgnaoui et al. found that two-week administration of probiotics was associated with decreased plasma adrenocorticotropic hormone (ACTH) and corticosterone concentration, and reduced hypothalamic corticotropin releasing factor (CRF) expression in response to stress [94]. They also found that the probiotic prevented colonic hyperpermeability, endotoxemia and central neuroinflammation [94]. As mentioned before, Brenner et al.’s systematic review also found positive results correlated with probiotic and prebiotic interventions, noting effects such as increased regulatory T cells, improved immunoregulation, and decreased stress and inflammation [14]. Finally, use of particular enteral antibiotics following TBI can counter dysbiosis and induce neuroprotection by increasing T_reg_ cell populations [95].

Several other targets are also currently under further study. Bansal et al. investigated vagal stimulation in TBI patients and discovered that this treatment promoted activity in enteric glial cells and limited intestinal permeability. Despite its administration in the setting of acute TBI not reducing mortality or unfavorable outcomes, progesterone mitigates injury in the gut by inhibiting proinflammatory cytokines [96,97]. While zonulin, a crucial protein in regulation of intestinal permeability, has not yet been widely explored as a treatment target for traumatic brain injury, one research team found that zonulin was involved in several diseases of autoimmune, neurodegenerative, and tumoral nature, and could function as an indicator of gut barrier damage in these conditions as well as a potential therapeutic target [98]. A recent review also discussed potential measures to reduce levels of zonulin and subsequently improve intestinal permeability [76]. This review cited a clinical study done in Austria that analyzed the effects of zeolite supplementation in endurance-trained individuals and found that this intervention correlated with decreased levels of excreted zonulin as well as decreased inflammation [76,99]. Stool zonulin was also found to be decreased in athletes who received a 20-day course of 500 mg of colostrum bovinum [76,100]. Probiotics were also implicated as a possible method in a colorectal cancer patient study that found a reduction in postoperative septicaemia and concentrations of serum zonulin following a daily oral probiotic mix [76,101]. While these studies have shown that zonulin levels can be decreased with various interventions, there is still limited outcome data, especially in regards to traumatic brain injury and the brain-gut axis.

Nutritional interventions are also being explored [102]. Recent research has shown that both gut microbial diversity and abundance vary between individuals based on several factors including diet [103]. One study further attributed diversity in the gut microbiome to number of unique plant species in a person’s diet. In this clinical investigation, McDonald et al. discovered that molecular alpha diversity was significantly increased in individuals with diets containing a high diversity of plants [104]. Furthermore, David et al. found that the microbial composition of the gut can be rapidly altered with dietary changes. This study placed human volunteers in either a plant-based diet group or an animal-based diet group, and the results showed a significant increase in diet-associated gut microbiota in the animal-based diet group. Particularly noteworthy is how quickly this type of change can take place in the gut microbiome—David et al. discussed that the alteration was noticeable within a day of placing the subjects on the animal-based diet, and original gut microbiomes were restored 2 days after concluding the diet [105]. This evidence would lead one to believe that more targeted diet changes could be a promising therapeutic avenue for conditions such as TBI that could potentially alter the gut microbiome.

In addition, in 2011, the Institute of Medicine recommended that patients with TBI should be given a high level of nutrition for two weeks to curtail inflammation [102]. Dietary treatments in the form of early enteral nutrition and intake of glutamine, arginine, nucleotides, and omega-3 fatty acids are another potential therapy that stimulates immune cells and promotes gut barrier health [106]. Vitamins and minerals such as nicotinamide, zinc, and magnesium have also shown potential in pre-clinical models [107].

## 6. Conclusions

Traumatic brain injury is a significant public health problem that can have devastating long-term consequences for patients including severe physical, cognitive, emotional, and/or behavioral disabilities. Clinical management of TBI focuses on prevention of secondary injury, but progression can be difficult to follow clinically. Through disruption of the BGA and the intimate involvement of the GI microbiota, TBI may initiate a feedback loop that potentiates a neuroinflammatory cascade and leads to secondary brain injury. TBI-induced dysbiosis, through its impact on the BGA, may potentiate secondary injury and influence functional outcome.

Relatively few studies have been conducted that focus on the impact of TBI on the gut microbiome, and the physiologic and clinical implications remain unseen. Many of the studies cited in our review are pre-clinical and involve animal models or cell culture, and it is to be seen whether these concepts will effectively translate into a clinical therapy. There is limited clinical data specifically regarding the effects of TBI on the microbiome and how the microbiome may then feed into that axis and further affect outcome.

We believe the relationship between TBI and the gut microbiome represents an area of study that could lead to a wide range of future research in new clinical intervention strategies. Alterations in the gut microbiome could potentially serve as a biomarker to improve detection of TBI and monitor progression or as a therapeutic target to prevent secondary injury in brain-injured patients. Therapeutic strategies such as FMT and probiotics may offer a neuroprotective benefit by targeting the dysregulated gut-microbiota-brain axis and restoring the gut microbiota to a healthier profile. Leveraging knowledge of the gut microbiome in the setting of TBI holds the exciting potential to influence treatment of brain-injured patients and enhance quality-of-life for patients with TBI.

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
