# Peer review of "A Review of Traumatic Brain Injury and the Gut Microbiome: Insights into Novel Mechanisms of Secondary Brain Injury and Promising Targets for Neuroprotection"

_brainsci, 2018, doi:10.3390/brainsci8060113_

Round 1

Reviewer 1 Report

Dear authors of the paper "A review of traumatic brain injury and the gut microbiome: Insights into novel mechanisms of secondary brain injury and promising targets for neuroprotection"

Please find below my comments relating to your paper. I believe there's much scope to improve the paper and hope you find feedback useful. 

---

Line 36: You claim that the "Physiological effects following TBI have been increasingly studied" yet the only reference provided is an animal study from 2009. As you set the expectation that the body of evidence on this subject is growing, the reader is bound to be disappointed by finding just 1 animal citation. 

Lines 36-40: The paper by Sundman is a narrative review. It's a good paper for an introduction section, but am unsure Kharrazian's paper provides any valuable insights. It's a review that lacks rigour and some of the papers included lack ecological validity. Perhaps this section would benefit from the robustness a couple of good quality systematic reviews on the subject instead. 

Lines 41-62: The literature cited is scarce and you don't make it absolutely clear that it is pre-clinical. I believe the reader would benefit from knowing that you're citing mostly animal studies. I would expect better signposting to make that absolutely clear, e.g. "based on animal models...". 

Additionally, you may have find the recent work by Brenner et al interesting: https://www.ncbi.nlm.nih.gov/pubmed/28606462. It's a systematic review including 3 good human studies that I believe you'll find valuable. 

Lines 62-124: Almost all references are reviews. Relying on citing reviews when reviewing a subject does a disservice to the reader as it ends up not providing much new knowledge. As your title includes the terms "novel" and "targets for neuroprotection" the reader will expect to find your synthesis of new content on this subject, which is lacking throughout this section.

Would it be possible to include some human clinical data to illustrate the concepts you're discussing in this section? Particularly when you say something like "numerous studies have demonstrated... " (line 103) but you proceed to cite only 1 study, this seems a little incongruous. 

Is your focus on clinical application or is it pre-clinical? I feel it's the former, butyour discussion is mostly pre-clinical in nature. In any event, the reader should benefit from your making that explicit. As the paper stands, it is unclear that the literature you're citing is mostly based on cell cultures and animal studies. 

Line 201: There are plenty of human studies that document the pathways you refer to in this section. Readers would benefit from you not repeating the same citations you used in the introduction. 

Lines 204-205: Wang et al's paper describes an animal model of reperfusion injury in rats but you're discussing the role of inflammation in the pathogenesis of Parkinsons, so the citation doesn't seem appropriate?

Perhaps you'd like to take a look at this recent review where you will find a number of references to clinical studies on this subject that you may want to include here, perhaps with a comment as to why they're relevant in the context you're exploring? https://www.ncbi.nlm.nih.gov/pubmed/29693607   

As mentioned previously, it is important that you make it clear for the reader to know when you're drawing from literature based on animal models and when you're discussing human / clinical data. Lines 203 to 205 cite 1 animal study and 1 review (citation number 61 - largely based on animal studies also). However on line 206 you say "Pre-clinical models have also demonstrated a relationship between gastrointestinal dysfunction and autism spectrum disorders, with a focus on the role of the BGA". It would be good to quote some human studies here (there are some) alongside animal models and to explain your rationale for choosing these papers?

Line 212: Study by Lee et al is an animal study. Please make this explicit. 

Lines 213-214:  What is the purpose of this phrase here? It seems a statement you'd make in an introduction, and the paper you cite is a generalist review. Perhaps you'd like to move this to the introduction section, or remove it altogether? Please review. 

Line 226: In line 226 you say: " Amongst patients with TBI, 8-33% of individuals exhibit autonomic imbalance [5]." You cite the review paper by Dr Kharrazian but the actual figures come from Kirk et al (2012) and refer to dysautonomia after pediatric brain injury (please see Dev Med Child Neurol. 2012 Aug;54(8):759-64. doi: 10.1111/j.1469-8749.2012.04322.x. https://www.ncbi.nlm.nih.gov/pubmed/22712762). The data you're presenting as applying to all patients with TBI actually refers to children only. Could you make this clear and get figures that apply to adults also? 

Rest of the paper, including "Potential Therapeutic Interventions". 

This section could provide a lot more insight into current potential therapeutic interventions. For example, you haven't mentioned microbial diversity or relative abundance and the fact that these can be altered by means of nutritional interventions. I would suggest you take a look at the recent research by McDonald et al: mSystems. 2018 May 15;3(3). pii: e00031-18. doi: 10.1128/mSystems.00031-18. eCollection 2018 May-Jun.

https://www.biorxiv.org/content/early/2018/03/07/277970. 

You mention zonulin. What about interventions to reduce zonulin levels, thereby improving intestinal permeability and reducing free radical load travelling to the brain via the vagus nerve? 

Overall, I feel the interventions section and the conclusion need strengthening and that doing this will have a  positive impact on how the paper flows. 

Author Response

The authors would like to thank the editor and reviewers of Brain Sciences for an extremely helpful and efficient review process. Several of the reviewers’ comments brought forth substantial limitations in the first version of the manuscript, which we have responded to directly. We appreciate the constructive comments and suggestions, and believe that addressing the reviewer comments has greatly improved the overall quality of the manuscript. Specific responses are below, and the revised version has been uploaded with changes highlighted, as requested.

Reviewers’ comments:

Reviewer #2: Dear authors of the paper "A review of traumatic brain injury and the gut microbiome: Insights into novel mechanisms of secondary brain injury and promising targets for neuroprotection"

Please find below my comments relating to your paper. I believe there's much scope to improve the paper and hope you find feedback useful. 

We thank you for your thorough review of our paper and we believe the changes we have made per your critique have greatly improved our work.

---

Line 36: You claim that the "Physiological effects following TBI have been increasingly studied" yet the only reference provided is an animal study from 2009. As you set the expectation that the body of evidence on this subject is growing, the reader is bound to be disappointed by finding just 1 animal citation. 

This was a very astute point brought up by the reviewer. This was an oversight on our part, and we have rectified the situation by attributing more references to this statement. (Line 36)

Lines 36-40: The paper by Sundman is a narrative review. It's a good paper for an introduction section, but am unsure Kharrazian's paper provides any valuable insights. It's a review that lacks rigour and some of the papers included lack ecological validity. Perhaps this section would benefit from the robustness a couple of good quality systematic reviews on the subject instead. 

Again, this is a great point. We agree that this section would benefit from a couple of systematic reviews. We have addressed the points made about Kharrazian’s paper by changing the citation to two papers, one of which is a systematic review (Lines 36-40). Later in this section, we also use another systematic review suggested in a later comment by reviewer #2 (Lines 55-59).

Lines 41-62: The literature cited is scarce and you don't make it absolutely clear that it is pre-clinical. I believe the reader would benefit from knowing that you're citing mostly animal studies. I would expect better signposting to make that absolutely clear, e.g. "based on animal models...". 

We thank the reviewer for bringing this to our attention. We have gone back through the entire paper and made it clear which studies are pre-clinical and which are clinical.

Additionally, you may have find the recent work by Brenner et al interesting:https://www.ncbi.nlm.nih.gov/pubmed/28606462. It's a systematic review including 3 good human studies that I believe you'll find valuable. 

We greatly appreciate the reviewer’s suggestion of the Brenner paper, which we found very helpful. We have included it in Lines 55-59.

Lines 62-124: Almost all references are reviews. Relying on citing reviews when reviewing a subject does a disservice to the reader as it ends up not providing much new knowledge. As your title includes the terms "novel" and "targets for neuroprotection" the reader will expect to find your synthesis of new content on this subject, which is lacking throughout this section.

We realize that this is a shortcoming in the original version of the manuscript that needed to be addressed and thank the reviewer for these suggestions to improve our paper. In as many places as possible, we have gone back and placed the original studies as the citation.

Would it be possible to include some human clinical data to illustrate the concepts you're discussing in this section? Particularly when you say something like "numerous studies have demonstrated... " (line 103) but you proceed to cite only 1 study, this seems a little incongruous. 

Another excellent point. We have addressed these comments by attributing more papers to the “numerous studies have demonstrated” part (Lines 107-108), and we have included more clinical studies in this section (Lines 124-127).

Is your focus on clinical application or is it pre-clinical? I feel it's the former, but your discussion is mostly pre-clinical in nature. In any event, the reader should benefit from your making that explicit. As the paper stands, it is unclear that the literature you're citing is mostly based on cell cultures and animal studies. 

This was definitely something worth reflecting on. While our focus is on clinical application, much of the existing research on the topics we cover are pre-clinical. However, we have made an effort to work in more clinical data (for instance, Lines 124-127), and we thank the reviewer for reminding us to focus our paper on the more clinical aspects.

Line 201: There are plenty of human studies that document the pathways you refer to in this section. Readers would benefit from you not repeating the same citations you used in the introduction. 

Another keen comment. We have changed the original reference to a human clinical study along with a couple of review papers that we believe will be of interest to readers. (Line 211)

Lines 204-205: Wang et al's paper describes an animal model of reperfusion injury in rats but you're discussing the role of inflammation in the pathogenesis of Parkinsons, so the citation doesn't seem appropriate?

Our apologies—the reference should be attributed to Houser et al. We have corrected this (Line 230).

Perhaps you'd like to take a look at this recent review where you will find a number of references to clinical studies on this subject that you may want to include here, perhaps with a comment as to why they're relevant in the context you're exploring?https://www.ncbi.nlm.nih.gov/pubmed/29693607   

We thank the reviewer for the excellent reference provided. This was a highly relevant paper, and we have referenced this study in this section, as well as later in our paper: Lines 237-239, Lines 340-348.

As mentioned previously, it is important that you make it clear for the reader to know when you're drawing from literature based on animal models and when you're discussing human / clinical data. Lines 203 to 205 cite 1 animal study and 1 review (citation number 61 - largely based on animal studies also). However on line 206 you say "Pre-clinical models have also demonstrated a relationship between gastrointestinal dysfunction and autism spectrum disorders, with a focus on the role of the BGA". It would be good to quote some human studies here (there are some) alongside animal models and to explain your rationale for choosing these papers?

Once again, we appreciate the reviewer bringing the need for clarification between pre-clinical and clinical studies. We have gone back through the entire paper and made it clear which studies are pre-clinical and which are clinical.

We have also made an effort to include multiple additional clinical studies to strengthen this section. (Lines 237-245).

Line 212: Study by Lee et al is an animal study. Please make this explicit. 

We thank the reviewer for catching this point, and we have made it clear that Lee’s study is an animal study (Line 248).

Lines 213-214:  What is the purpose of this phrase here? It seems a statement you'd make in an introduction, and the paper you cite is a generalist review. Perhaps you'd like to move this to the introduction section, or remove it altogether? Please review. 

We thank the reviewer for bringing this to our attention, and we agree that this was a paragraph that needed work. We intended for this paragraph to discuss how the brain-gut axis could be disrupted by autoimmune or inflammatory mechanisms. We have clarified this by adjusting the first sentence of this paragraph (Line 246). We have also edited our wording to make it clear that Lee’s paper is a pre-clinical study, and we expanded our discussion of Lee’s paper to improve the logical flow of this paragraph (Lines 248-255).

Line 226: In line 226 you say: " Amongst patients with TBI, 8-33% of individuals exhibit autonomic imbalance [5]." You cite the review paper by Dr Kharrazian but the actual figures come from Kirk et al (2012) and refer to dysautonomia after pediatric brain injury (please see Dev Med Child Neurol. 2012 Aug;54(8):759-64. doi: 10.1111/j.1469-8749.2012.04322.x.https://www.ncbi.nlm.nih.gov/pubmed/22712762). The data you're presenting as applying to all patients with TBI actually refers to children only. Could you make this clear and get figures that apply to adults also? 

We appreciate the reviewer’s analysis of this section. We have reviewed the Kirk paper noted by the reviewer as well as several other papers. We have altered this citation to include the Baguley study we found on dysautonomia as well as a few other papers that corroborate the statistic of 8-33% adult patients with TBI who also suffer from dysautonomia. We apologize for the initial confusion and have made it explicit that we are discussing adult patients (Line 271).

Rest of the paper, including "Potential Therapeutic Interventions". 

This section could provide a lot more insight into current potential therapeutic interventions. For example, you haven't mentioned microbial diversity or relative abundance and the fact that these can be altered by means of nutritional interventions. I would suggest you take a look at the recent research by McDonald et al: mSystems. 2018 May 15;3(3). pii: e00031-18. doi: 10.1128/mSystems.00031-18. eCollection 2018 May-Jun.

https://www.biorxiv.org/content/early/2018/03/07/277970. 

These are some excellent points made by the reviewer, and we greatly appreciate the paper recommended by the reviewer. We have included further detail about microbial diversity, relative abundance, and nutritional interventions. (Lines 351-364). We have also incorporated the findings of the McDonald paper suggested by the reviewer (Lines 354-356)

You mention zonulin. What about interventions to reduce zonulin levels, thereby improving intestinal permeability and reducing free radical load travelling to the brain via the vagus nerve? 

We thank the reviewer for this insightful comment about zonulin. We have made an effort to discuss potential zonulin interventions (Lines 336-350).

Overall, I feel the interventions section and the conclusion need strengthening and that doing this will have a  positive impact on how the paper flows. 

We thank the reviewer for the constructive critique. We have greatly expanded upon our “Potential Therapeutic Interventions” section and strengthened our conclusion.

Reviewer 2 Report

This is a review paper attempting to link PB with TBI as a therapy which is a plausible approach.

It is quite well written although some sections had repetitive statements which seemed redundant.  The microbiome and the brain and the systemic injury headings seem like they could be combined in a more succinct discussion.

Overall I thought the paper was sufficiently interesting for the readership of the journal

Author Response

The authors would like to thank the editor and reviewers of Brain Sciences for an extremely helpful and efficient review process. Several of the reviewers’ comments brought forth substantial limitations in the first version of the manuscript, which we have responded to directly. We appreciate the constructive comments and suggestions, and believe that addressing the reviewer comments has greatly improved the overall quality of the manuscript. Specific responses are below, and the revised version has been uploaded with changes highlighted, as requested.

Reviewers’ comments:

Reviewer #1: This is a review paper attempting to link PB with TBI as a therapy which is a plausible approach.

We appreciate the reviewer’s accurate summary.

It is quite well written although some sections had repetitive statements which seemed redundant.  The microbiome and the brain and the systemic injury headings seem like they could be combined in a more succinct discussion.

This is an excellent point brought up by the reviewer. We appreciate the suggestion to improve the flow of the manuscript, and we addressed this point by restructuring our paper. We deleted the original “Gut Microbiome and the Brain” section, and instead incorporated the information from those paragraphs into the “Brain-Gut Axis & Neurologic Injury” section (Lines 210-234).

Overall I thought the paper was sufficiently interesting for the readership of the journal

We thank the reviewer for the encouraging comments.

Round 2

Reviewer 1 Report

Dear Author,

Thank you so much for your feedback on my comments to version 1 of your manuscript and for taking into consideration many of my recommendations. I feel the paper flows much better and provides the reader with a much better understanding of the involvement of the brain-gut axis in neurologic injury. This is a very exciting area of work and your paper highlights the potential application to the improvement of TBI treatment in humans.